# Post-Traumatic Atlanto-Axial Instability: A Combined Clinical and Radiological Approach for the Diagnosis of Pathological Rotational Movement in the Upper Cervical Spine

**DOI:** 10.3390/jcm12041469

**Published:** 2023-02-12

**Authors:** Bertel Rune Kaale, Tony J. McArthur, Maria H. Barbosa, Michael D. Freeman

**Affiliations:** 1Firda Medical Center AS, 6823 Sandane, Norway; 2CAPHRI School for Public Health and Primary Care, Faculty of Health, Medicine, and Life Sciences, Maastricht University, 6211 LM Maastricht, The Netherlands

**Keywords:** upper cervical instability, atlas-axis rotational test (A-ART), CT scan, whiplash trauma

## Abstract

Post-traumatic rotational instability at the atlanto-axial (C1-2) joint is difficult to assess, much less quantify, due to the orientation and motion plane of the joint. Prior investigations have demonstrated that a dynamic axial CT scan, during which the patient maximally rotates the head right and left, can be used to evaluate and quantify the amount of residual overlap between the inferior articulating facet of C1 and the superior facet of C2, as an index of ligamentous laxity at the joint. We have previously demonstrated that a novel orthopedic test of rotational instability, the atlas-axis rotational test (A-ART), may have utility in identifying patients with imaging evidence of upper cervical ligament injury. In the present investigation, we assessed the correlation between a positive A-ART and a CT scan assessment of the relative quantity of residual C1-2 overlap, as a percent of the superior articulating facet surface area of C2. A retrospective review was conducted of the records of consecutive patients presenting to a physical therapy and rehabilitation clinic, over a 5-year period (2015–20) for chronic head and neck pain after whiplash trauma. The primary inclusion criteria were that the patient had undergone both a clinical evaluation with A-ART and a dynamic axial CT to evaluate for C1-2 residual facet overlap at maximum rotation. The records for a total of 57 patients (44 female/13 male) were identified who fit the selection criteria, and among these, there were 43 with a positive A-ART (i.e., “cases”) and 14 with a negative A-ART (i.e., “controls). The analysis demonstrated that a positive A-ART was highly predictive of decreased residual C1-2 facet overlap: the average overlap area among the cases was approximately one-third that of the control group (on the left, 10.7% versus 29.1%, and 13.6% versus 31.0% on the right). These results suggest that a positive A-ART is a reliable indicator of underlying rotational instability at C1-2 in patients with chronic head and neck symptoms following whiplash trauma.

## 1. Introduction

Intervertebral instability secondary to intervertebral ligamentous injury is a relatively common finding among patients with whiplash trauma-related chronic neck pain [1]. The clinical presentation can be particularly complicated when the instability is in the upper cervical spine (i.e., between the occiput, atlas, and axis [C0-2]), as patients may suffer from nonspecific symptoms of headache, vertigo, and neck pain, the origin of which can be difficult to pinpoint [2].

The diagnosis of symptomatic spinal instability requires a combination of symptoms consistent with the condition, and radiographic evidence of extra-physiologic movement at the joint in the relevant plane and direction, which may or may not be accompanied by MRI evidence of ligamentous disruption. A diagnosis of anterior or posterior instability of the sub-axial spine (C2–C7) may be made via flexion and extension radiographs, or more involved evaluations of wider range of movements can be accomplished via fluoroscopic examination [1]. In contrast, the evaluation and diagnosis of instability in the upper cervical spine is made more difficult by the anatomical complexity and predominant type of movement of the joints, which is rotation about the vertical axis [3]. Thus, while lateral flexion instability of C1 on C2 can be evaluated dynamically with anterior to posterior open mouth radiographs, excessive rotational movement can only be evaluated via imaging in an axial (up to down) orientation. 

There are two prior studies that have described the use of CT scanning to assess the degree of rotational movement at C1 on C2 by quantifying the loss of facet joint surface overlap at maximum voluntary head rotation. The first study, from 1999, evaluated an uninjured population of 10 children, and described a maximal joint contact loss of 74 to 85% [4]. The second study, published 10 years later, evaluated the percent decrease of joint overlap in a healthy group of 40 adults at maximal head rotation, finding an average loss of 70% of joint overlap (range 42–86%), thus leaving an average residual overlap area of 30% [5]. No other publications describing an investigation of the technique in either healthy or injured populations were identified, following a search of the literature using key terms.

In 2008, Kaale and colleagues described a novel clinical examination protocol for evaluating upper cervical rotational instability (UCRI) called the “atlas-axis rotational test,” abbreviated as “A-ART” henceforth [6]. The orthopedic test is performed on a passive seated patient, by palpating and stabilizing the transverse process of C2 while rotating the patient’s head and palpating the degree of end play of the lateral mass of C1 at maximal tolerable rotation, see Figure 1. Instability of C1 on C2 is graded 0–3 based on the perceived degree of abnormality of end play. The authors compared the A-ART results of 122 patients to MRI evaluation of the integrity of the upper cervical ligaments (alar and transverse) and tectorial and posterior atlanto-occipital membranes. When the clinical test results were dichotomized as either normal (0–1) or abnormal (2–3), there was good to excellent agreement (i.e., kappa coefficient of 0.7–0.9) between the ability of the A-ART rotational test to detect abnormal joint end play and the MRI confirmation of ligamentous abnormality. Although these results were encouraging, they could not confirm that the instability inferred from the dynamic clinical examination was in fact correlated with actual instability, as the MRI evaluations were performed in a neutral position, and thus cannot be considered a “gold standard” test of rotational instability of C1 on C2. The validation of A-ART for the detection of UCRI would have utility in the medicolegal investigation of the pain source in patients with chronic symptoms suggestive of upper cervical instability following whiplash trauma, as the test could help identify patients with a higher likelihood of positive objective imaging indicative of traumatic injury. 

In this study we present the results of an investigation of the correlation between the results of a dynamic orthopedic test for UCRI (the A-ART) and a dynamic imaging analysis of UCRI, via residual C1-C2 facet overlap analysis. The purpose of the investigation is to assess the diagnostic accuracy of the A-ART using a direct radiographic measure of excessive joint excursion and instability. 

## 2. Methods

### 2.1. Inclusion Criteria

The data used for the analysis were retrospectively abstracted from medical records and imaging files for patients who had been referred by their general practitioner (GP) to a single physiotherapy and rehabilitation practice for evaluation and treatment of chronic post-traumatic neck pain (range 4 to 8 years after injury), from 2015 through 2020. The primary inclusion criteria for the study were that (1) there was a relatively high clinical suspicion of UCRI based on the clinical presentation, and (2) both the A-ART and a dynamic rotational CT scan were performed on the patient. 

The clinical suspicion of UCRI was based on the presence of chronic (i.e., >6 months duration) neck pain complaints combined with symptoms potentially of a craniocervical origin, including dizziness, headache, and a sense of head pressure, including a worsening of the symptoms with head rotation, including during normal activities. The provoked symptoms in some cases would persist for hours to days. The A-ART was performed on the patients by 2 blinded clinicians, and graded 0 to 3. and the patient was subsequently referred to an outpatient imaging center for a CT scan of the upper cervical spine which included a dynamic rotational stress protocol. As noted above, prior to the CT scan, all of the patients had a cervical MRI study in order to rule out significant CNS or musculoskeletal pathology.

A total of 57 patients (44 female/13 male) were identified for study, after the exclusion of one patient with a suspected connective tissue disorder. For the purposes of the study, most accurately described as a prospective cohort design, patients were dichotomized into 2 groups by A-ART grade, following the same protocol described by Kaale et al. [6], in which a result of 2–3 was deemed “abnormal” or positive for UCRI (little to no stop feeling of C1 lateral mass at the end of rotation), and 0–1 was deemed “normal” and negative for UCRI (solid or soft stop feeling during rotation). See Figure 1. Two physiotherapist examiners, both experienced with application of the A-ART, had to agree that the A-ART grade was 2 or 3 for the patient to be categorized in the “abnormal” group. As these data were gathered retrospectively, no protocol was in place to blind the second examiner to the first examiner’s finding, and thus the level of inter-examiner agreement between initial findings could not be reconstructed. For further analysis, the patients with an abnormal A-ART result were deemed as “cases” and the patients with a normal A-ART test were deemed “controls”.

### 2.2. CT Scan Protocol

Referral for the CT scan was provided by the referring GP, and was based on either a high degree of suspicion of UCRI among the 43 patients with a positive A-ART, or to rule out other cervical spine pathology (including UCRI) in the 14 A-ART negative patients. The CT scan was performed on the same day (and at the same facility) as a cervical MRI study, which was ordered at the same time at the CT scan. All patients were provided with information regarding the risks of the procedure, and given the alternative to opt out of the diagnostic study as part of the procedures, alternatives, and risk (PAR) conference. The scans were obtained from just above the base of the skull to the T1/T2 level, and performed on a Toshiba Aquilion ONE CT scanner, using 80 kVp and 80 mAs, with a 0.5 s scan time. Bone and soft tissue target algorithms were used, and the scans were performed without gantry angulation. The scans were 0.5 mm thick and were obtained in one single volume of 160 mm and reconstructed as 0.5mm axial slices every 0.25 mm, yielding a total of 320 × 0.25 mm axial slices. The 2 rotational scan sequences were each approximately 4 min in duration.

The entire cervical spine scan was performed with the patient’s head in a neutral position, and then upper cervical images were obtained with the head in maximal tolerable rotation, so as to reproduce the conditions of the A-ART in a supine position, using previously described upper cervical imaging protocol [7,8]. 3D Volumetric CT scans were reviewed on a Vitrea (Vital Images) workstation using both 3D and cross-sectional imaging techniques. The total scan dose was 2.2 mSv (millisieverts). For reference, the doses of an abdominal CT scan and single chest X-ray are 10 and 0.02 mSv, respectively [9].

### 2.3. CT Scan Interpretation

The neutral position scans were first evaluated for significant pathology, which were negative for all patients. To evaluate the atlas-axis facet coverage at maximal rotation, the axial slices that optimized the view of the cortical rim around the facets was used. The joint surface of the superior facet of C2 was then identified, and the online software program GeoGebra Classic (http://www.geogebra.org/) was used to delineate the anatomical perimeter of the articulating surface, as well as quantify the area [10]. Next, the posteromedial margin of only the part of the inferior articulating surface of C1 that was overlapping with the C2 superior facet surface was outlined, and the area of overlap was calculated by the software as a percentage of the area of the C2 joint surface, see Figure 2a–c.

### 2.4. Statistical Analysis

Welch’s Two-Sample t-test was used to analyze the differences in the mean values of right and left overlap percentage, as well as the distribution of age between the case and control groups. The multivariate relationships between right and left overlap and age between the 2 groups were analyzed using generalized linear modeling. The Kolmogorov–Smirnov (K-S) test of normality was used to assess the distribution of the ages in each group. A *p*-value of 0.05 or lower was considered statistically significant for all analyses, which were performed using RStudio, version 2022.020 + 443 (RStudio Team: Integrated Development for R. RStudio, PBC, Boston, MA, USA).

### 2.5. Consent

All patients were contacted and asked for consent for their anonymized archived medical information to be used for the present investigation. All patients gave consent. This study was exempted from ethics review because of the use of archived medical information, which was described collectively, rather than individually.

## 3. Results

Both right and left overlap percentages were significantly lower among the patients with an abnormal A-ART, versus the patients with normal tests (see Table 1). The average percentage of overlap among the cases was approximately one-third of the average in the control group (10.7% versus 29.1% on the left, and 13.6% versus 31.0% on the right). Although the right side values were slightly higher than the left side values in both groups, the difference was not significant. While normally distributed in both groups (i.e., K-S was not significant), age was significantly lower in the patient group; on average, the cases were 8.1 years younger than the control group, with a range of ages among the cases of 15 to 70, and 28 to 77 for the controls. There was a nominal disparity in sex distribution between cases and controls, with 10/43 (23%) males in the former, and 3/14 (21%) males in the latter. Multivariate linear regression was used to examine the role of age and sex as a predictor of overlap; however, no significant associations were discerned.

## 4. Discussion

The results of the present investigation are noteworthy in two respects; they demonstrate the potential utility of axial CT scanning for the objective evaluation of rotational instability in patients with symptoms consistent with upper cervical instability, and they provide quantitative objective evidence of the clinical utility of the A-ART for identifying rotational ligamentous instability at C1-2. Both findings are unique in the literature. Prior investigations of upper cervical rotational instability have primarily focused on the evaluation of injury to the alar ligaments.

Despite the fact that the 14 subjects in the control group were seeking treatment for persisting craniocervical symptoms following a cervical spine trauma (i.e., whiplash trauma), the average residual atlanto-axial overlap in the control group of approximately 29 to 31% (left and right, respectively) fell well within the range described by Mönckeberg and colleagues in an asymptomatic population of 40 adults (30%) [5]. In contrast, the residual overlap among the cases in the present investigation fell outside the range observed in the prior asymptomatic population study, in which the lowest residual overlap of any of the subjects was 14.3%, versus the 10.7% and 13.6% (left and right, respectively) average among the cases.

Aside from significantly lower residual overlap at C1-2, the distinguishing feature among the cases, versus the controls, was an abnormal A-ART. It is thus reasonable to infer from these findings that the A-ART is a relatively accurate test for atlanto-axial rotational instability, although the degree of accuracy cannot be quantified from these data, as there is no gold standard threshold or cut point to measure the individual findings against. While the findings may also be attributable to an unexamined confounding factor related both to instability and the A-ART (aside from sex and age, which were not found to be correlated in the analysis), this explanation is unlikely, given that the pathomechanics resulting in the decreased C1-2 facet overlap would also reasonably result in a palpable alteration in joint end play (and abnormal A-ART result). The authors have not observed any negative effects during administration of the A-ART; however, we advise a slow and cautious approach any time maximal head rotation is assessed in the patient with suspected ligamentous instability in the upper cervical spine, keeping in mind the proximity of the bony structures to the upper cervical spinal cord.

As noted in Section 2, all of the patients underwent an upper cervical CT scan on the same day that they underwent a cervical MRI study. A look back at the MRI images revealed that the majority did not include the C1-2 levels, and thus the correlation between the CT scan evidence of instability with possible MRI evidence of ligamentous integrity was not feasible given the limitations of the available data. This may be a fertile avenue of future investigation, however. Based on the results of the present study, it is reasonable that for the patient with persistent symptoms of upper cervical instability and a positive (i.e., Grade 2+) A-ART, that the next step in evaluating the source of the ongoing symptoms would include both a cervical MRI and the CT scan of upper cervical rotation.

Prior investigations of upper cervical ligamentous injury have largely focused on non-dynamic imaging of morphological changes in the upper cervical ligaments, with particular focus on the alar and transverse ligaments, with the head in a neutral position [11,12]. The degree of association between such imaging findings and patient outcomes, if any, is uncertain, however [13]. Dynamic CT evaluations of the upper cervical spine have been described in the evaluation of instability secondary to rheumatoid arthritis or healing odontoid fractures, but few describe the technique for the evaluation of ligamentous integrity following traumatic injury [14,15]. In one such study, authors used CT scanning to quantify rotational instability in 47 patients with chronic symptoms after whiplash trauma, in comparison with 26 uninjured controls, by quantifying the relative segmental rotation at each level versus total cervical rotation [16]. The authors found excessive rotation at C0-C1 in the injured group, but not at C1-2. In contrast with the present study, however, no prior research has evaluated residual overlap at C1-2 in an injured population. 

There are several potential weaknesses of the present investigation which prompts caution in interpreting the results: the foremost is that the A-ART evaluation was performed by two examiners with substantial experience with the test, and thus, the reliability of the test when performed by other clinicians cannot be established with these results. Further, there are no established norms for the amount of palpable rotational movement between C1 and C2 that would fall into the negative (Grade 0–1) versus positive (Grade 2–3) A-ART result, and the ~5 mm threshold described for the test is an unmeasured approximation. Additionally, the average and range of values for the residual facet overlap evaluation in healthy adults was only found in a single prior publication; thus the technique is relatively novel and has not been validated on a more diverse population. The results of the present study are the first to describe residual facet overlap in a symptomatic population with a history of traumatic injury, however.

For patients with rotational instability at C1-2, therapeutic options are minimal, and results uncertain. Some, but not all of the patients in the current study had positive results from rehabilitation and physical therapy modalities. For patients with refractory symptoms, surgical fusion of C1-C2 is a viable option, although outcomes and complication rates are not well established.

We cautiously interpret these results to suggest that the atlas-axis rotational test (A-ART) is a potentially useful physical examination tool for identifying the pathological source of persisting cervicocranial symptoms consistent with upper cervical rotational instability. As such, the test offers potential benefits in the medicolegal investigation of the pain generator in patients with persisting unexplained upper cervicocranial symptoms after whiplash trauma, in that the patients most likely to have objective imaging evidence of rotational instability at C1-2 can be identified with greater accuracy, thus providing legally admissible proof of the location and extent of injury. Further investigation is warranted to evaluate the practicality and diagnostic accuracy of the A-ART in larger patient populations.

## 5. Conclusions

An accurate diagnosis can be elusive for the patient with persisting whiplash trauma-related craniocervical symptoms, in part because injury that results in rotational instability can be difficult to identify or quantify. The use of the atlas-axis rotational (A-ART) orthopedic test may provide reliable evidence for the presence of upper cervical instability, and should be considered as a useful initial test for differential diagnosis of the source of persisting head and neck symptoms in patients with chronic pain following whiplash trauma. For patients with a positive A-ART, an axial CT scan will provide a definitive diagnosis, as well as quantification, of rotational instability.

## Figures and Tables

**Figure 1 jcm-12-01469-f001:**
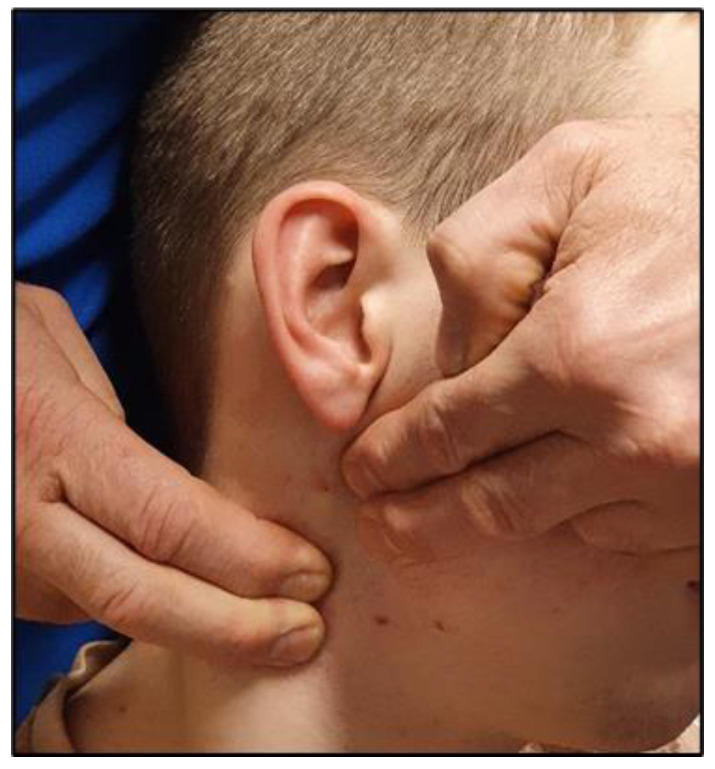
Examiner hand position during the atlas-axis rotational test (A-ART). While standing behind the seated patient, the examiner places both hands on the occipito-cervical junction, opposite the side of head rotation. With the 2nd and 3rd fingers, the examiner’s lower hand (right, in the photograph) is used to stabilize and traction posteriorly against the transverse process of C2. The 2nd and 3rd fingers of the other hand (left, in the photograph) contact the mastoid process of the occiput and lateral mass of the atlas, respectively. The test is then performed with varying angles of cervical rotation, to locate the position that yields maximal movement between C1 and C2, and graded by the amount of C1 versus C2 movement described in the text. For the purposes of the present study, a grade of 0–1 equates to little perceived relative rotational motion between the transverse processes of C1 and C2 (subjectively gauged, less than ~5 mm), and 2 or more exceeds this threshold.

**Figure 2 jcm-12-01469-f002:**
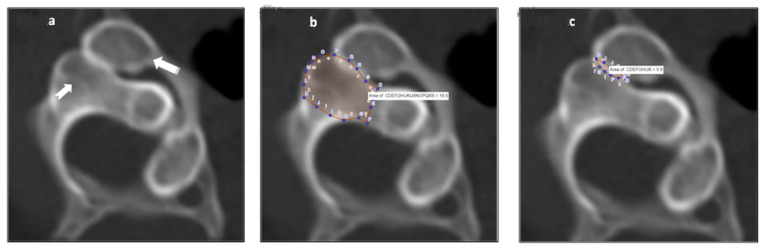
(**a**) CT scan, axial view, with the patient’s head rotated maximally to the left. The notched arrow points to the right superior articulating facet of C2, and the striped arrow points to the right inferior articulating facet of C1. (**b**) The same scan as in Figure 2a, with the outer margin of the right superior articulating facet of C2 outlined using the GeoGebra software, resulting in an area of 18.9 cm^2^. (**c**) The same scan as depicted in Figure 2a and b, but with the residual overlapping area of C1 and C2 outlined using the GeoGebra software, resulting in an area of 0.9 cm^2^. The area of residual overlap, as a percent of the total area of the superior articulating facet of C2, is calculated as (0.9/18.9 × 100%), and is thus 4.8%. (R—right, L—left, A—anterior, P—posterior).

**Table 1 jcm-12-01469-t001:** Mean Values (Standard Deviations, SD) of percent of atlanto-axial residual facet overlap at the extreme of right and left rotation, and average difference between the 2 groups (95% confidence intervals [CI]), and age distribution among 43 cases and 14 controls.

	Mean Values (SD)				
	N	Female/Male	R Overlap	L Overlap	Age (Years)
Cases	43	33/10	13.6% (5.81)	10.7% (5.06)	39.8 (12.27) Range 15–70
Controls	14	11/3	31.0% (10.11)	29.1% (11.15)	47.9 (12.83) Range 28–77
*p*-value			<0.001	<0.001	0.0499
Mean difference (95% CI)			17.4% (11.5, 23.3)	18.4% (12.0, 24.8)	8.1 years (0.02, 16.2)

## Data Availability

The data that served as the source material for the analysis are provided in a Appendix A.

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
