# Peer review of "Post-Traumatic Atlanto-Axial Instability: A Combined Clinical and Radiological Approach for the Diagnosis of Pathological Rotational Movement in the Upper Cervical Spine"

_jcm, 2023, doi:10.3390/jcm12041469_

Round 1
Reviewer 1 Report
The article is interesting and well written.
Is it possible to define a cut off for the C1C2 overlap? Is it possible to add some information about the MRI of the included patients? The presence of ligaments injury is, to my knowledge the most used criterion for suspicion of rotational upper cervical rotational instability.
Please add a paragraph about possible therapy for post traumatic rotational instability.
Are the authors able to suggest a diagnostic workflow for this condition? For example MRI and then CT if clinical and A-ART positive?
In Authors' experience has the A-ART maneuver any risk of cord injury in non-expert hands? Is there a rate of false- positive or false-negative results?
Author Response
We are very appreciative of the reviewer’s comments, and have done our best to adjust the paper to reflect them, as follows (our changes are in bold []s after the comments):
Reviewer 1:
The article is interesting and well written.
Is it possible to define a cut off for the C1C2 overlap? Is it possible to add some information about the MRI of the included patients? The presence of ligaments injury is, to my knowledge the most used criterion for suspicion of rotational upper cervical rotational instability.
[at line 200 we added the following, with regard to a cut point:
It is thus reasonable to infer from these findings that the A-ART is a relatively accurate test for atlantoaxial rotational instability, although the degree of accuracy cannot be quantified from these data, as there is no gold standard threshold or cut point to measure the individual findings against.
At line 112:
The CT scan was performed on the same day (and at the same facility) as a cervical MRI study, which was ordered at the same time at the CT scan.
And, at line 208:
As noted in the methods section, all of the patients underwent an upper cervical CT scan on the same day that they underwent a cervical MRI study. A look back at these images revealed that the majority did not include the C1-2 levels, and thus the correlation between the CT scan evidence of instability with possible MRI evidence of ligamentous integrity was not feasible given the limitations of the available data. This may be a fertile avenue of future investigation, however.]
Please add a paragraph about possible therapy for post traumatic rotational instability.
[Added to line 215:
For patients with rotational instability at C1-2, therapeutic options are minimal, and results uncertain. Some, but not all of the patients in the current study had positive results from rehabilitation and physical therapy modalities. For patients with refractory symptoms, surgical fusion of C1-C2 is a viable option, although outcomes and complication rates are not well established.]
Are the authors able to suggest a diagnostic workflow for this condition? For example MRI and then CT if clinical and A-ART positive?
[we have added the following to line 212:
Based on the results of the present study, it is reasonable that for the patient with persisting symptoms of upper cervical instability and a positive (i.e., Grade 2+) A-ART, that the next step in evaluating the source of the ongoing symptoms would include both a cervical MRI and the CT scan of upper cervical rotation.]
In Authors' experience has the A-ART maneuver any risk of cord injury in non-expert hands? Is there a rate of false- positive or false-negative results?
[At line 207 we added the following:
The authors have not observed any negative effects during administration of the A-ART, however, we advise a slow and cautious approach any time maximal head rotation is assessed in the patient with suspected ligamentous instability in the upper cervical spine, keeping in mind the proximity of the bony structures to the upper cervical spinal cord.]
Reviewer 2 Report
Thank you for giving me the opportunity to review this interesting and important work.
· Line 83: Please state the design of this study clearly?
· Line 105: Please provide some details on the 2 testers; was level of agreement established, level of experience?
· Line 180: your aim says “In this study we present the results of an investigation of the correlation between the results of a dynamic orthopedic test for UCRI (the A-ART) and a dynamic imaging analysis of UCRI, via residual C1-C2 facet overlap analysis. The purpose of the investigation is to assess the diagnostic accuracy of the A-ART using a direct radiographic measure of excessive joint excursion and instability”. However, although you do use regression to look at predictors, it is not clear how you present the correlation between tests and imaging y, or diagnostic accuracy evaluation [sensitivity and specificity] in your analysis.
· It is not clear why you chose to use sex and age as predictors?
· There is no comment on sample size.
Author Response
Reviewer 2:
We are very appreciative of the reviewer’s comments, and have done our best to adjust the paper to reflect them, as follows (our changes are in bold []s after the comments):
- Line 83: Please state the design of this study clearly?
[At line 84 the following was added:
For the purposes of the study, most accurately described as a prospective cohort design, patients were dichotomized into 2 groups by A-ART grade, patients were dichotomized into 2 groups by A-ART grade, following the same protocol described by Kaale et al]
Line 105: Please provide some details on the 2 testers; was level of agreement established, level of experience?
[At line 88, the following was added:
Two physiotherapist examiners, both experienced with application of the A-ART, had to agree that the A-ART grade was 2 or 3 for the patient to be categorized in the “abnormal” group. As these data were gathered retrospectively, no protocol was in place to blind the first examiner to the second examiner’s finding, and thus the level of inter-examiner agreement between initial findings could not be reconstructed.
- Line 180: your aim says “In this study we present the results of an investigation of the correlation between the results of a dynamic orthopedic test for UCRI (the A-ART) and a dynamic imaging analysis of UCRI, via residual C1-C2 facet overlap analysis. The purpose of the investigation is to assess the diagnostic accuracy of the A-ART using a direct radiographic measure of excessive joint excursion and instability”. However, although you do use regression to look at predictors, it is not clear how you present the correlation between tests and imaging y, or diagnostic accuracy evaluation [sensitivity and specificity] in your analysis.
[At line 200 the following discussion was added:
Aside from significantly lower residual overlap at C1-2, the distinguishing feature among the cases, versus the controls, was an abnormal A-ART. It is thus reasonable to infer from these findings that the A-ART is a relatively accurate test for atlantoaxial rotational instability, although the degree of accuracy cannot be quantified from these data, as there is no gold standard threshold or cut point to measure the individual findings against.]
- It is not clear why you chose to use sex and age as predictors?
[At line 173, the discussion of the results was modified to:
Although normally distributed in both groups (i.e., K-S was not significant), age was significantly lower in the patient group; on average, the cases were 8.1 years younger than the control group, with a range of ages among the cases of 15 to 70, and 28 to 77 for the controls. There was a nominal disparity in sex distribution between cases and controls, with 10/43 (23%) males in the former, and 3/14 (21%) males in the latter. Multivariate linear regression was used to examine the role of age and sex as a predictor of overlap, however, no significant associations were discerned.]
- There is no comment on sample size.
[The effect of sample size is, in part, reflected by the tests of statistical significance.
We did add, at line 231, the following:
Further investigation is warranted to evaluate the practicality and diagnostic accuracy of the A-ART in larger patient populations.]
Reviewer 3 Report
The authors present an interesting study on atlanto-axial instability with a well-structured analysis of the radiological data. Here my comments:
1. The atlanto-axial instability should be better defined as a clinical disease both with signs, symptoms and radiological criteria;
2. The A-ART should be better described, especially depicting when the test is positive and negative with quantitative and qualitative parameters; “Instability of C1 on C2 is graded 0-3 based on the perceived degree of abnormality of end play” is too generic; grades should be extensively described in a dedicated table;
3. Thresholds of overlapping percentages distinguishing normal articulating facets from abnormal ones should be specified together with the criteria supporting them;
4. The range of ages in both groups appears too wide (15-70, 28-77) considering bias related to the arthrosis of the facet joints. The youngest or the oldest patients should be excluded or the authors should explain how the different degrees of C1-C2 rotation among ages have been somehow standardized;
5. The correlation between the A-ART and the dynamic CT-scan in identifying the rotational ligamentous instability should be analyzed in a statistical way providing even a graph about that;
6. Taking in account that one of the core messages of the paper is the adoption of a dynamic CT scan for a cranio-cervical junction pathology, the authors should mention other papers adopting “dynamic” CT scan for diseases in that anatomical region such as the paper for conservative treatment of C2 fractures on ESJ 2019, May with PMID: 30673876.
I would consider the manuscript for publication after a careful revision of the above mentioned critical aspects.
Author Response
Reviewer 3.
We are very appreciative of the reviewer’s comments, and have done our best to adjust the paper to reflect them, as follows (our changes are in bold after the comments):
- The atlanto-axial instability should be better defined as a clinical disease both with signs, symptoms and radiological criteria;
At line 38, the following was added:
The diagnosis of symptomatic spinal instability requires a combination of symptoms consistent with the condition, and radiographic evidence of extra-physiologic movement at the joint in the relevant plane and direction, which may or may not be accompanied by MRI evidence of ligamentous disruption.
At line 77, the following was added:
The clinical suspicion of UCRI was based on the presence of chronic (i.e., >6 months duration) neck pain complaints combined with symptoms potentially of a craniocervical origin, including dizziness, headache, and a sense of head pressure, including a worsening of the symptoms with head rotation, including during normal activities.
- The A-ART should be better described, especially depicting when the test is positive and negative with quantitative and qualitative parameters; “Instability of C1 on C2 is graded 0-3 based on the perceived degree of abnormality of end play” is too generic; grades should be extensively described in a dedicated table;
At line 98, the following was modified:
Examiner hand position during the atlas-axis rotational test (A-ART). While standing behind the seated patient, the examiner places both hands on the occipito-cervical junction, opposite the side of head rotation. With the 2nd and 3rd fingers, the examiner’s lower hand (right, in the photograph) is used to stabilize and traction posteriorly against the transverse process of C2. The 2nd and 3rdfingers of the other hand (left, in the photograph) contact the mastoid process of the occiput and lateral mass of the atlas, respectively. The test is then performed with varying angles of cervical rotation, to locate the position that yields maximal movement between C1 and C2, and graded by the amount of C1 versus C2 movement described in the text. For the purposes of the present study, a grade of 0-1 equates to little perceived relative rotational motion between the transverse processes of C1 and C2 (subjectively gauged, less than ~5 mm), and 2 or more exceeds this threshold.
Also added at line 235:
…the A-ART evaluation was performed by 2 examiners with substantial experience with the test, and thus, the reliability of the test when performed by other clinicians cannot be established with these results. Further, there are no established norms for the amount of palpable rotational movement between C1 and C2 that would fall into the negative (Grade 0-1) versus positive (Grade 2-3) A-ART result, and the ~5 mm threshold described for the test is an unmeasured approximation
- Thresholds of overlapping percentages distinguishing normal articulating facets from abnormal ones should be specified together with the criteria supporting them;
[at line 200 we added the following, with regard to a cut point:
It is thus reasonable to infer from these findings that the A-ART is a relatively accurate test for atlantoaxial rotational instability, although the degree of accuracy cannot be quantified from these data, as there is no gold standard threshold or cut point to measure the individual findings against.
- The range of ages in both groups appears too wide (15-70, 28-77) considering bias related to the arthrosis of the facet joints. The youngest or the oldest patients should be excluded or the authors should explain how the different degrees of C1-C2 rotation among ages have been somehow standardized;
At line 178 the following was added:
While normally distributed in both groups (i.e., K-S was not significant), age was significantly lower in the patient group; on average, the cases were 8.1 years younger than the control group, with a range of ages among the cases of 15 to 70, and 28 to 77 for the controls. There was a nominal disparity in sex distribution between cases and controls, with 10/43 (23%) males in the former, and 3/14 (21%) males in the latter. Multivariate linear regression was used to examine the role of age and sex as a predictor of overlap, however, no significant associations were discerned.
- The correlation between the A-ART and the dynamic CT-scan in identifying the rotational ligamentous instability should be analyzed in a statistical way providing even a graph about that;
The t-test results provided in table 1 give the statistical basis for the conclusions that a positive A-ART test is predictive of lower facet overlap. This statistic isn’t a correlation, so there is no graphical way to illustrate it, however.
|
|
Mean values (SD) |
||||
|
|
N |
Female/Male
|
R overlap |
L overlap |
Age (years) |
|
Cases |
43 |
33/10 |
13.6% (5.81) |
10.7% (5.06) |
39.8 (12.27) Range 15-70 |
|
Controls |
14 |
11/3 |
31.0% (10.11) |
29.1% (11.15) |
47.9 (12.83) Range 28-77 |
|
p-value
|
|
|
< 0.001 |
< 0.001
|
0.0499 |
|
Mean difference (95% CI) |
|
|
17.4% (11.5, 23.3)
|
18.4% (12.0, 24.8) |
8.1 years (0.02, 16.2) |
- Taking in account that one of the core messages of the paper is the adoption of a dynamic CT scan for a cranio-cervical junction pathology, the authors should mention other papers adopting “dynamic” CT scan for diseases in that anatomical region such as the paper for conservative treatment of C2 fractures on ESJ 2019, May with PMID: 30673876.
Thank you for the excellent suggestion. We have added the following to the discussion section, line 235:
Dynamic CT evaluations of the upper cervical spine have been described in the evaluation of instability secondary to rheumatoid arthritis or healing odontoid fractures, but few describe the technique for the evaluation of ligamentous integrity following traumatic injury.[14,15] In one such study, authors used CT scanning to quantify rotational instability in 47 patients with chronic symptoms after whiplash trauma, in comparison with 26 uninjured controls, by quantifying the relative segmental rotation at each level versus total cervical rotation.[16] The authors found excessive rotation at C0-C1 in the injured group, but not at C1-2. In contrast with the present study, however, no prior research has evaluated residual overlap at C1-2 in an injured population.
Round 2
Reviewer 3 Report
After a minor spell check I think the manuscript is now suitable for publication